# Assessing Motivations and Channels for Nutritional Information Verification in Spanish University Communities

**DOI:** 10.3390/ijerph22030357

**Published:** 2025-02-28

**Authors:** Paula Von-Polheim, Carolina Moreno-Castro

**Affiliations:** Research Institute on Social Welfare Policy (POLIBIENESTAR), University of Valencia, 46022 Valencia, Spain; carolina.moreno@uv.es

**Keywords:** food communication, eating habits, self-checker, science communication, diets, food security

## Abstract

This research analyses the results obtained from a survey performed on the nutritional and eating habits of Spanish university communities (students, faculty and administrative staff), involving four multiple-choice questions which determine the respondents’ level of interest in dietary topics. The study sample comprised 124 respondents from three Spanish universities: Complutense University of Madrid, University of Valencia and University of Malaga. The statistical software program R was used to conduct both analyses (quantitative and qualitative), using frequencies and percentages for the multiple-choice questions. The main results reveal that the respondents’ level of interest and motivations were decisive when deciding on whether to verify nutritional information or not. The use of different verification platforms underscores the urgent need for reliable sources and educational intervention in nutrition fields.

## 1. How Information Shapes Eating Habits: The Growing Interest in Nutrition

The growing interest in nutritional information shown by academia and the general public has been driven by a combination of factors, including health concerns, personal well-being and the influence of digital media, social networks and other online resources, which have become critical tools for food education [1,2,3,4,5]. In their analysis of the role of technology in the greater desire to consume, Kozinets et al. concluded that public and professional networks ‘drive consumption passion’ by creating a new universe of desire intensified by technological interfaces [6]. This transformation in consumption habits, particularly the positive role of technology in promoting healthy eating habits [7], offers a hopeful outlook for nutrition.

This is evidenced by the findings of many exploratory studies of how university students and other population groups actively seek nutritional information. However, they often face difficulties in assessing the quality and credibility of information sources [8,9]. In this context, people’s eating habits, especially those of university students, are strongly influenced by the information they receive and how they perceive it. Regarding this aspect, other researchers have highlighted generational differences in the perception and use of nutritional information [8,10,11,12]. Younger students tend to adopt a hedonistic approach to eating, focusing on the physical appearance of food and immediate pleasure. In contrast, young adults place more value on overall health and long-term well-being [10]. This trend is reflected in preferences for certain types of foods, such as whole-grain or low-calorie products, depending on the kind of information they receive or actively seek [8,11].

These findings stress how important it is to investigate the information channel preferences of consumers to gain a better understanding of this state of affairs. According to Blázquez et al., 2018 and Zhang, 2012 [9,13], even though social media and online platforms are the most frequently used sources, they often provide inconsistent and sometimes misleading information about food quality [2,13]. In contrast, the use of more reliable sources, such as scientific journals and academic databases, leads to more informed and healthier dietary decisions [2,9]. This preference for readily accessible but less dependable information may be linked to a lack of critical digital literacy skills, which hampers the ability of students to distinguish between reliable and unreliable sources [14]. Against this backdrop, digital literacy and trustworthy information sources are both critical factors influencing how consumers interpret and act upon nutritional information [14,15,16].

### 1.1. Contextual, Demographic and Cultural Factors Influencing Eating Habits

In light of the literature, contextual, demographic and cultural factors have a profound influence on eating habits and nutritional information channel preferences. Shafiee et al. have recently highlighted the importance of incorporating traditional foods and cultural practices into health promotion strategies in their research on the perceptions of healthy eating among urban Indigenous populations in Saskatchewan [17]. This culturally sensitive approach is also evident in the study performed by Palascha and Chang, 2024 [18], examining how messages about healthy and sustainable eating resonate better with consumers of lower socioeconomic status, underscoring the need to tailor health messages to the target audience’s characteristics.

Moreover, research suggests that food preferences and the reception of nutritional information can vary significantly across countries and cultures, thus pointing to the need to adapt interventions and communication strategies to specific cultural contexts [19,20]. To be more effective, nutritional information campaigns should consider cultural values and food beliefs [21].

The powerful influence of community education campaigns and community-based interventions can impact food choices and improve nutritional literacy. These interventions are more effective when designed with the specific needs and characteristics of the target communities in mind [22,23]. However, some studies, such as that performed by Fernández Torres et al., 2014 [24], suggest that these interventions may be insufficient for bringing about sustained changes in eating habits, especially when structural factors influencing food choices, such as availability and pricing, are not addressed.

Research conducted in various dining settings, such as university canteens and cafés, has shown that consumers value the price–quality relationship, food naturalness and the clarity of available nutritional information [25]. However, simplified labels, such as the Nutriscore system, are more effective in facilitating healthy food choices, particularly in contexts where consumers have little time to decide [13,26].

In this regard, Cohen and Babey (2012) highlight that eating behaviours are primarily automatic in response to contextual stimuli and are influenced by the visual appearance of products, such as portion sizes and labelling [27]. Although labelling partially affects food choices, its effectiveness depends on the context and type of information provided, suggesting the need for more personalised interventions according to the environment [28]. Regarding consumer preferences, De-Magistris et al., analysed how European consumers valued public food labels more than private ones, with the ‘Denomination of Origin’ label being the most appreciated [29]. This finding is consistent with the results obtained by Méjean et al., 2013 [30], who identified patterns in the perception of front-of-pack labels based on socioeconomic characteristics and nutritional knowledge. Both studies suggest that demographic factors can have a significant influence on labelling preferences, indicating the need to design communication strategies tailored to different consumer segments.

In their analysis of channel strategies on digital platforms, Zhou et al., underscored the need to consider multiple influencing factors, from the product to the market level, which is particularly relevant in food labelling [31]. Shangguan et al. revealed that food labelling can reduce calorie intake and improve the food choices of consumers. However, there are still several research gaps [32]. For instance, despite extensive studies on labelling preferences and the effects of educational interventions, there is still little understanding of how different levels of personal motivation affect the long-term adoption of healthy eating behaviours [33].

### 1.2. The Influence of Information Sources and Channels on Eating Habits

In recent decades, there have been numerous enquiries into the effectiveness of educational interventions in promoting healthy eating habits. For intake, studies such as those performed by Báti, 2024, Evenhuis et al., 2018, Wang et al., 2022 and Weingarten et al., 2022 [34,35,36,37] have assessed intervention programmes in school and university canteens and cafés, finding that those solely based on providing information are insufficient for prompting significant changes in eating habits [34]. It has been observed that the most effective interventions are those combining the provision of information with menu redesign or changes in the food environment to facilitate the choice of healthy foods [9,35].

Besides the environment, the role of influencers and social media has also been an object of study. As a result of their research on how young people in Hong Kong consumed dietary information provided by social media celebrities, Wong et al. concluded that this information had a more significant impact on men than on women, while its effectiveness depended on the credibility of such celebrities [38]. This is in keeping with the study performed by Dam et al. [39] on how parasocial interactions with mukbang influencers [40] affect consumption intentions, highlighting the importance of perceived credibility in the effectiveness of food marketing on social media. Focusing on the United Kingdom, Goodman and Jaworska stressed the importance of understanding the scope of action of opinion leaders in the digital sphere and how they engaged their audience, before emphasising the need to characterise these mainstream discourses to understand their subsequent implications for user preferences, such as changes in eating habits [41].

In this digital context, emerging forms of communication, such as augmented reality technologies and intelligent systems, have demonstrated their potential for promoting balanced eating practices. In their study of the use of augmented reality technologies in nutritional education, Chanlin and Chan demonstrated that students using an augmented reality system improved their dietary knowledge and behaviour, while underscoring the potential of technology to foster healthy eating habits [42]. Other studies have highlighted the role of recipe-sharing platforms in creating communities that share food wellness practices, plus their ability to persuade people to change their eating habits and styles [31,43]. Many people become involved in social media and online communities to pursue personal health goals, such as healthy eating and increased physical activity. ‘However, people struggle with impression management and reaching the right audience when sharing health information on these platforms’ [44] (p. 1674).

Regarding attitudes towards new food processing techniques, Pivarnik et al. revealed that members of the fresh produce and seafood industry and food safety educators, alike, had a limited knowledge of non-thermal processing methods, with 50 and 67 per cent of the respondents, respectively, scoring below the 80 per cent mastery threshold [45]. The lack of knowledge and ambivalent attitudes towards these technologies underline the need for effective educational programmes aimed at promoting the adoption of innovative methods that ensure food safety and quality.

### 1.3. Digital Information Verification and Its Impact on Eating Behaviour

Bell and Marshall developed and validated a Food Involvement Scale (FIS), which proved to be a reliable measure for assessing individual differences in the level of interest in food [46]. According to this scale, their findings indicated that individuals with higher food involvement had a greater ability to discriminate between food samples with different sensory characteristics, such as sweetness, acidity, saltiness and fat content. The authors went on to underscore the importance of food involvement as a mediator in food choice behaviour, while also pointing to its relevance for research on food behaviour and sensory testing.

Regarding the consumption of products designed to enhance cognitive performance, Egreja et al. analysed online information-sharing practices among students and how they resorted to forums and blogs to discuss the use of drugs and natural supplements [47]. The results showed that most of them turned to these digital spaces to share experiences and seek advice, emphasising the critical role of the Internet as a source of information on health topics. However, they also highlighted the fact that they had difficulty in discerning the reliability of information, thus increasing their exposure to potential risks. This phenomenon is related to the study performed by Denniss et al., 2024 [48], which found that 34.8 per cent of nutrition posts on influential Instagram accounts in Australia were of low quality, whereas only 6.1 per cent were considered to be of high quality. This study suggests that nutritional information on social media can be misleading and underscores the need to improve the quality and trust of online dietary communication.

For their part, Teunissen et al. developed and validated the Food Media Content Gratifications Scale (FMCG), a tool for measuring the gratifications obtained from food-related media content [49]. This seven-factor scale (entertainment, social connections, ‘food porn’, information on food cultures, health and cooking convenience) shows that the consumption of food media content follows patterns similar to those of other media content. The validity and reliability of the scale make it an essential tool for investigating how food media content influences eating attitudes and related behaviours, thus supplementing the findings of Bell and Marshall on food involvement and its effects on food choice behaviour [46].

Addressing the credibility of information and its impact on the evaluation of innovative food products, Walten and Wiedmann demonstrated that independent sources, such as food scientists, were perceived as more credible than in-house product developers, resulting in a more favourable product evaluation. Information overload can, however, reduce behavioural intention, as a Scheffé post hoc test revealed a significant difference between high and low information conditions [50]. This finding is in line with the results obtained by Mladenovic et al., 2024 [51], who found that the perceived credibility of ‘green’ information on food products significantly influenced consumer sustainability evaluations, highlighting the importance of presenting credible information sources to improve the perception of sustainable products.

### 1.4. Effective Communication Strategies for Enhancing the Public Understanding of Science Information

Research has stressed the importance of using effective communication methods, such as visual tools, to enhance the understanding and acceptance of science topics on the part of the public by taking into consideration their individual and contextual characteristics [52]. Specifically, infographics have been shown to improve memory retention and attitude change towards genetically modified foods [53]. In this vein, initiatives aimed at enhancing the public understanding of science information should consider the audience’s cultural context, especially as regards polarising issues such as genetically modified foods [54]. Studies have revealed that individuals, particularly those with higher education, are aware of the factors affecting the acquisition of scientific knowledge and recognise that perceptions of experts and authorities vary according to the educational level [55]. Most attribute the differences in opinion among experts to the general difficulty in obtaining scientific knowledge and the interests involved rather than to differences in the excellence or background of those experts.

In addition, some studies have highlighted the growing interest in countering health misinformation, one of which performed by Chen and Tang examining how threat perception and self-affirmation influence the willingness of experts to correct online misinformation [56]. The results show that self-affirmation and messages posing a high health threat or placing the accent on high efficacy significantly increase this willingness, suggesting the need for specific strategies to encourage healthcare professionals to correct false information. This approach concurs with the findings of Fähnrich et al., 2023 [57], stressing the importance of establishing quality criteria for online science communication, particularly in contexts where digital platforms are the primary means of dissemination. According to Lee and Lee [53], furthermore, the consumption of science information through digital media can positively influence the public perception of genetically modified foods, provided that the information is presented in an accessible and contextualised manner.

Regarding food risk perception and its relationship with consumption decisions, studies conducted in Italy [58] and the United Kingdom [59] suggest that concerns about food safety, such as the risks associated with genetically modified foods or additives, can significantly influence consumer choices. This finding underscores the importance of improving food risk communication by providing clear, evidence-based information.

In conclusion, the present pilot study began with the findings of international authors who stated that interest in nutrition positively correlates with the likelihood of verifying information. In addition, personal motivations, such as dietary care and healthcare, significantly influence this verification tendency, with health-related concerns often prompting individuals to rely on more credible sources like scientific or academic journals. On the other hand, the selection of verification platforms would be subject to variation, with general media and YouTube being the most cited sources in other international studies. However, the reliability of these platforms could vary depending on their positioning relative to other sources.

This state-of-the-art analysis suggests a more comprehensive approach tailored to diverse realities. Therefore, this study aims to explore the nutritional and dietary information channels used by university community members and assess whether they verify the information they receive to ensure its accuracy and prevent misinformation.

In order to meet this objective, the following research questions were formulated:
(i)What is the relationship between health-related motivations and the preference for more reliable information sources?(ii)How do personal interests and specific motivations influence the verification of nutritional information?(iii)How can promoting accessible and reliable sources improve nutritional information-seeking practices?

To conduct this research, a questionnaire with four multiple-choice questions was designed to analyse the level of interest of the respondents in dietary topics. The study sample comprised 124 respondents from three different Spanish universities: the Complutense University of Madrid, the University of Valencia and the University of Málaga. The selection criteria employed were aimed at creating a sample that was academically and geographically diverse.

## 2. Methods

This research was designed as a pilot study to understand how university communities verified food and nutrition information. We worked with a small but focused sample of 124 respondents from three large universities, which allowed us to gather early insights into their verification behaviours. Since pilot studies are meant to test the items, multiple choices and questions before launching larger research projects, using a smaller sample made sense at this stage, according to Petticrew and Roberts, 2006 [60]. Our survey goal was to check if this line of research had the potential for a more extensive, nationwide study in the future.

### 2.1. Participants

Participants were self-selected through a short survey distributed via LimeSurvey, an online platform used by the University of Valencia. The survey, available in Spanish and English, included an informed consent document (see Appendix A) and was completed by 124 respondents from the Complutense University of Madrid, the University of Valencia, and the University of Malaga. These universities were chosen to represent Spain’s diverse academic communities and geographical regions. We presented the survey during different academic outreach activities as a pilot study for future national project implementation. The LimeSurvey platform ensured that all data collection, storage, and processing complied with the General Data Protection Regulation (GDPR) and Spanish data protection laws, offering anonymisation options, explicit consent mechanisms, and secure encryption to maintain high ethical and privacy standards.

### 2.2. Data Collection

Data collection was carried out between April and May 2022, based on a questionnaire consisting of one open-ended and four multiple-choice questions, administered via mobile devices (see the complete questionnaire in Appendix A). The multiple-choice questions were designed to measure the respondents’ interest in food-related topics, whereas the aim of the open-ended question was to identify the methods they used to cross-check the nutritional information they received. They provided detailed replies that allowed for assessing the reliability of the information they encountered and the different methods employed to verify its accuracy.

### 2.3. Analysis Techniques

The research employed two types of analysis:
(a)A quantitative analysis to measure, in percentage terms, the respondents’ level of interest in food-related topics and how many of them verified nutritional and dietary information.(b)A qualitative descriptive analysis of the respondents’ replies to gain a deeper understanding of how they accessed, evaluated and verified nutritional information and to identify potential areas for improvement in education on information verification and healthy eating practices.

The statistical software R Version 0.0.3 (R Core Team, Vienna, Austria, 2022) [61] was used to conduct both analyses. The data collected were summarised using frequencies and percentages for the multiple-choice questions, while a qualitative study of the open-ended responses was performed to identify common information verification patterns and strategies.

## 3. Results

### 3.1. Results of the Quantitative Analysis

#### Statistical Analysis

Table 1 shows the relationship between the respondents’ interest in nutritional and eating habits and in verifying information relating to these topics. Of the 27 respondents who admitted having little interest in nutrition, only 19 per cent verified the information they encountered. In contrast, a significantly higher proportion of the 97 respondents (56%) who expressed an interest in nutrition made an effort to verify such information.

This notable disparity underscores the crucial role that personal interest plays in verifying nutritional information. Individuals who demonstrate a greater interest in nutrition are more inclined to seek accurate and reliable information, suggesting that motivation and engagement with the topic are key drivers in fostering critical evaluation practices. Consequently, enhancing interest in nutrition could be an essential strategy for promoting more rigorous information verification behaviours among the public at large.

Table 2 illustrates the impact of different reasons for verifying nutritional information. It reveals that the respondents with an interest in dietary care and those actively following a diet were the most likely to engage in verifying the information they encountered, with 62 and 57 per cent, respectively, reporting verification behaviours. In contrast, those respondents with past experiences of food intoxication were the least likely to verify nutritional information, with none of them (0%) in this category engaging in such practices.

The variation in these percentages highlights the influence of personal reasons and past experiences on attitudes towards information verification. It suggests that motivations rooted in proactive health management or current dietary practices make individuals more inclined to verify nutritional information than those with negative past experiences.

Furthermore, Table 3 also provides insights into the platforms used to verify information on nutrition and eating habits. General media and YouTube were the most frequently used sources, with 61 and 56 per of the respondents, respectively, relying on these channels. Interestingly, Facebook was a unique case in that all the respondents (100%) using the platform to this end verified the information they encountered on it.

These findings suggest that the accessibility and perceived credibility of different platforms play a significant role in whether users verify the information they encounter there or not. They also imply that while highly accessible mainstream platforms are widely used, those perceived as more trustworthy, like Facebook, may foster more consistent user verification practices.

### 3.2. Results of the Qualitative Descriptive Analysis

#### Principal Component Analysis (PCA)

Principal component analysis was used to identify patterns in the data and reduce their dimensionality.

Screenplot 1 (Figure 1) illustrates the percentage of variance explained by each dimension in the dataset. In this screenplot, as there is no evident point of change or ‘elbow’, this suggests a uniform variance distribution across the dimensions. This uniformity indicates that each dimension contributes similarly to explain the overall variance within the dataset. For instance, when considering variables such as the use of Instagram, Twitter and Facebook for information verification, the graph does not reveal any dominant factor or small group of factors that disproportionately accounts for the respondents’ behaviour when verifying nutritional information. This lack of a clear point of inflexion suggests that no single dimension or subset of dimensions has an overwhelming influence on explaining the data’s variance, thus pointing to a complex, multi-faceted pattern in relation to how individuals engage with different platforms for verifying nutritional content.

However, Screenplot 2 presents a disparate picture, for there is a noticeable point of change starting from the second dimension, indicating that the first two principal components capture a significant proportion of the variance in the dataset. This observed ‘elbow’ suggests that these two dimensions effectively summarise the critical patterns in the survey responses. The first dimension could represent a general interest in nutrition and health, encompassing broader attitudes towards these topics. In contrast, the second dimension conceivably reflects more specific variables, such as the channels used for information verification, whether these be social media platforms like Instagram or more authoritative sources like scientific journals.

The differences between the two screenplots provide valuable insights. In Screenplot 1, the absence of a clear point of change indicates that the respondents’ verification behaviours were not affected by any specific factor, reflecting a diverse range of influences. Conversely, Screenplot 2 suggests that the first two dimensions play a critical role in capturing the main variations in the data, implying that these dimensions encompass the most relevant factors driving the respondents’ verification practices.

In conclusion, the analysis of these two screenplots in Figure 1 reveals different patterns in variance distribution. While the first screenplot shows a uniform contribution across all dimensions, the second one highlights the significant explanatory power of the first two dimensions, emphasising their importance in summarising the main patterns within the data.

The biplot represented in Figure 2 emphasises the importance of different variables through their squared cosine values, which indicate the contribution of each variable to defining the dimensions. This is because variables with higher squared cosine values are more influential in shaping these dimensions. For instance, if the variable ‘interest in the origin of food’ has a high squared cosine value, this signifies that it is a critical factor in understanding the primary dimension, highlighting its strong association with how the respondents engaged with nutritional information.

Overall, the combined analyses of the screenplots and biplot offer a detailed insight into the patterns of interaction between university community members and nutritional information. The first two dimensions identified in the screenplots are particularly effective for capturing the primary trends and variations in the data, suggesting that these dimensions represent the most critical aspects of the respondents’ information-seeking behaviours. For its part, the biplot provides a nuanced view of the specific relationships between the respondents and the diverse variables influencing their verification practices, such as their personal motivations, preferred platforms and interest in particular nutritional topics.

Together, these visual tools create a comprehensive framework for understanding the complexities of the information verification behaviours of university community members, highlighting key factors that shape their engagement with nutritional content.

## 4. Discussion

The analysis of the survey data reveals significant trends in how university community members obtain information about nutrition and eating habits. Basically, personal interest and specific motivations play a crucial role in these practices. Additionally, the choice of verification platforms varies significantly, suggesting the need to promote reliable and accessible sources to improve the quality of the nutritional information available to students.

This pilot study builds a solid foundation for understanding attitudes and behaviours relating to nutritional information in the university context, while also highlighting potential areas for educational and communication interventions.

With respect to the research questions, the survey findings highlight three remarkable aspects:
(a)Interest and information verification. There is a positive correlation between interest in nutritional topics and the likelihood of verifying information. This is evident in the clustering variables relating to personal interest and active information verification in the biplot, as revealed in a previous study performed by Moreno et al. [62].(b)Diverse motivations. Personal motivations, such as dietary care and healthcare, significantly influence the tendency to verify information. The biplot shows how these motivations are related to specific channels, with the health-related kind often being associated with more reliable sources like scientific or academic journals.(c)Preferred platforms. The respondents’ choice of platforms for information verification varied. For instance, the positioning of general media and YouTube, albeit popular, in the biplot is relative to other variables.

The expanding role of digital media in disseminating nutritional information has revolutionised how individuals, particularly students, access and interpret dietary recommendations. For Casino and Rabassa [1] and Moreno et al. [3], digital platforms have become central to food education. However, the varying quality of information on these platforms, as highlighted by Blázquez et al. [13] and Gazibara et al. [2], poses a challenge to students who may not possess the critical skills necessary to differentiate between reliable and unreliable sources.

Our study corroborates these concerns, for it shows that the respondents with a genuine interest in nutrition were more inclined to verify information in this respect. This finding, which is consistent with the results obtained by Zhang [9], suggests that interest in a topic fosters a more rigorous approach to information verification. In addition, those who are motivated by health concerns tend to gravitate towards more credible sources, such as scientific journals, thereby enhancing the quality of the information they consume. Conversely, those with less intrinsic motivation may rely on less credible social media platforms, which often fail to meet the rigorous standards of scientific validity. This disparity in source preference underscores the necessity for enhanced digital literacy programmes. As contended by Horgan and Sweeney [14], critically assessing information sources is vital in our era of misinformation. Therefore, educational initiatives should focus on equipping students with the skills for critically evaluating the credibility of different information sources.

Our findings also highlight the fact that university community members resort to a variety of verification platforms. The preference for general media and popular platforms like YouTube points to a broader issue of accessibility versus reliability. While these platforms are readily accessible, their content often lacks the scientific rigour required for making informed dietary decisions. This variability in platform preference indicates a need for more accessible and reliable nutritional information sources.

Educational interventions combining information dissemination with changes in the food environment appear to be more effective. According to Evenhuis et al. [35] and Zhang [9], providing information is insufficient for changing eating behaviours. Interventions that also involve modifying the food environment—such as redesigning menus or offering healthier food options—are likely to have a significant impact on the eating habits of university community members.

The influence of social media and emerging technologies on nutritional education should not be overlooked, like, for example, influencers and augmented reality technologies which could enhance dietary knowledge and practices [38,42]. In this connection, using these innovative technology’s opinion leaders, particularly those with a high perceived credibility, can have a significant impact on dietary choices, for they offer novel ways of engaging young people with a view to improving their nutritional knowledge. However, the effectiveness of these tools depends on content credibility and the appropriateness of the platforms and social apps [41]. It is essential to underscore that influencers are expected to play an increasingly more important role in health communication in the coming years, this being ‘mainly due to the targeted demand of predominantly underage users and the high attractiveness of influencer marketing from a commercial point of view’ [63] (p. 8).

Moreover, our results emphasise the importance of contextual, demographic and cultural factors in shaping eating behaviours and nutritional information channel preferences. Research has shown the need for culturally sensitive health messages [17,18], for tailoring them to the cultural and socioeconomic profile of the target audience is crucial for effective communication. This approach ensures that nutritional advice is relevant to and resonates with diverse population segments.

The data collected also show that community-based education campaigns play a critical role in influencing food choices. Interventions designed with community-specific needs in mind can enhance nutritional literacy and promote healthier eating habits habits [22,23]. Nevertheless, these interventions must address structural factors like food availability and pricing, which can significantly impact eating behaviours [24].

The effectiveness of food labelling systems, such as Nutriscore, in facilitating healthy food choices highlights the role of simple, clear information in decision-making. While labelling can influence food choices, its impact depends on the context and type of information provided [13,26]. Simplified labels can be particularly effective when consumers need to make quick decisions.

## 5. Conclusions

Our research has helped to gain a much better understanding of how nutritional information is sought, verified and processed by university community members. In particular, it underscores the importance of designing tailored educational and communication strategies for addressing the unique needs and behaviours of university students, faculty and administrative staff. Recognising the diverse personal motivations, platform preferences and specific contexts of these institutions, future research should focus on refining nutritional education and information dissemination methods. This would enable more effective interventions aimed at enhancing the accuracy and accessibility of dietary information.

Given that students, faculty and administrative staff spend a great deal of time on university campuses, due to the nature of academic, research and administrative responsibilities, the challenge of maintaining healthy dietary practices is plain to see. University canteens and cafés play a pivotal role in shaping these nutritional habits, which is why university leadership must ensure that the food served is inclusive, diverse and of high nutritional quality. Menu options should cater to a heterogeneous population with different dietary needs and preferences, while also reflecting cultural and nutritional inclusivity and promoting healthier behaviours across the board.

Moreover, the pervasive issue of misinformation in the digital age poses a significant barrier to healthy dietary behaviours. Combating misinformation in the university setting is not only a matter of providing factual nutritional information but also ensuring that this information is readily available, credible and contextually relevant. It is incumbent upon university administrations, health educators and researchers to foster an environment that promotes health literacy and equips students and staff with the tools to evaluate critically formal and informal nutritional information sources.

In light of the growing body of evidence linking diet to academic performance, mental well-being and long-term health outcomes [64,65], this focus on improving nutritional literacy within university communities is timely and essential.

Future lines of research should explore more interdisciplinary approaches, integrating public health, education and digital communication insights to craft comprehensive strategies that tackle these challenges. Such initiatives will undoubtedly contribute to the university community’s health, well-being and academic success.

## Figures and Tables

**Figure 1 ijerph-22-00357-f001:**
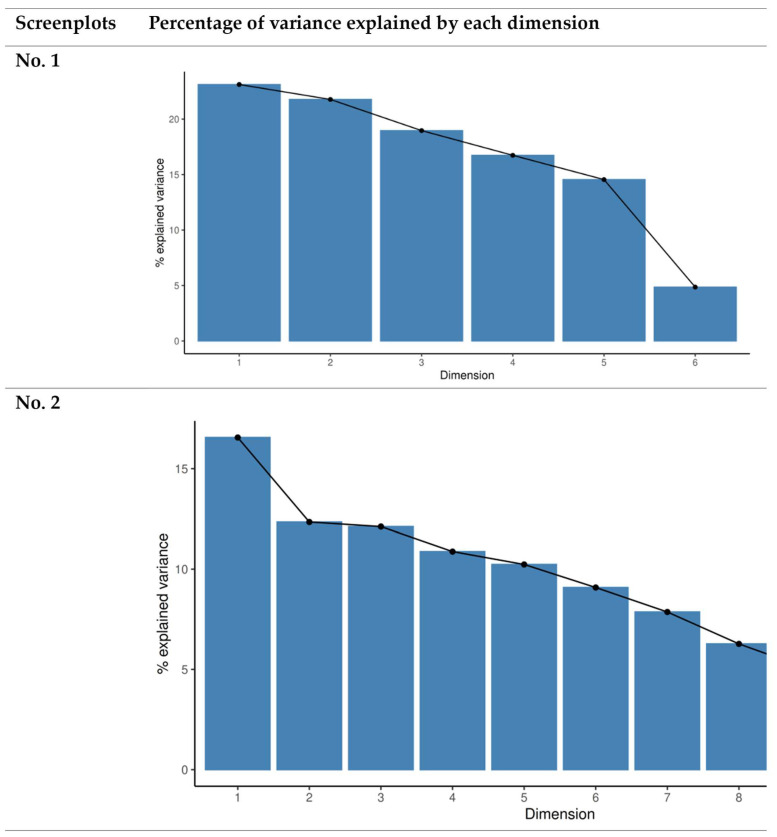
Screenplots showing the relevant patterns.

**Figure 2 ijerph-22-00357-f002:**
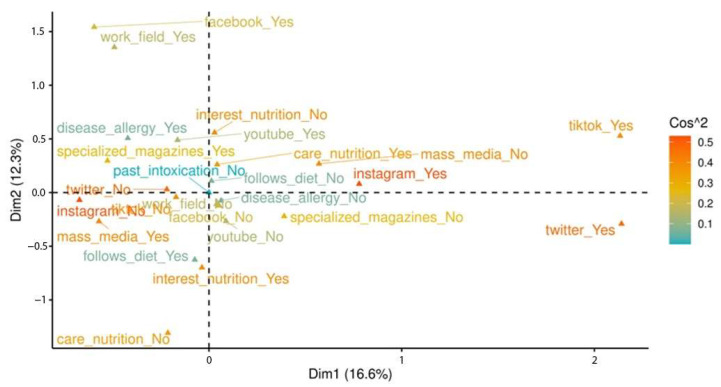
Biplot for the first two dimensions (28.9% variance explained).

**Table 1 ijerph-22-00357-t001:** Information verification by interest in nutritional topics.

Cross-Checking Information on Nutrition and Food	Interest in Nutritional and Food Topics	
	No, n = 27 ^1^	Yes, n = 97 ^1^
No	22 (81%)	43 (44%)
Yes	5 (19%)	54 (56%)

Source: own elaboration. ^1^ n (%).

**Table 2 ijerph-22-00357-t002:** Information verification by motivations.

	Cross-Checking Information	
	No, n = 43 ^1^	Yes, n = 54 ^1^
Dietary care	28 (38%)	45 (62%)
Work or study in that field	5 (56%)	4 (44%)
Following or willing to follow a diet	6 (43%)	8 (57%)
Interest in the origin and preparation of food	24 (50%)	24 (50%)
Suffering from a disease or some type of allergy that requires it	9 (56%)	7 (44%)
Past intoxication	1 (100%)	0 (0%)
Other	1 (17%)	5 (83%)

Source: own elaboration. ^1^ n (% per row).

**Table 3 ijerph-22-00357-t003:** Platforms and media used for information verification.

	Information Verification	
	No, n = 43 ^1^	Yes, n = 54 ^1^
Instagram	27 (52%)	25 (48%)
Twitter	6 (55%)	5 (45%)
Facebook	0 (0%)	4 (100%)
TikTok	4 (50%)	4 (50%)
YouTube	15 (44%)	19 (56%)
General print/digital media	17 (39%)	27 (61%)
Specialised printed/digital magazines on related topics	21 (48%)	23 (52%)
Other	3 (21%)	11 (79%)

Source: own elaboration. ^1^ n (% per row).

## Data Availability

The original contributions presented in this study are included in the article and Appendix A. Further inquiries can be directed to the corresponding author.

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
