# Peer review of "Assessing Motivations and Channels for Nutritional Information Verification in Spanish University Communities"

_ijerph, 2025, doi:10.3390/ijerph22030357_

Round 1
Reviewer 1 Report
Comments and Suggestions for Authors
Please see file attached.

Author Response
The responses could be found in the manuscript written in red colour
Comment 1:
The abstract should be rewritten, beginning with a background paragraph. As it stands, it presents the study without demonstrating the relevance of university communities' search for nutritional information. It is focused on presenting the methodology, sample and results without presenting the research question or objectives. I would recommend revising this text so that the reader can better understand its purpose. The expression ‘health’ is absent, and it might be beneficial to include it in the initial presentation.
Response 1:
Thank you so much for the suggestion. If editor agree with it, we tried to modify and amply the abstract. Sometimes, it depends on the editorial policy about the abstract words.
Comment 2:
Introduction: there is no connection between the end of the introduction and the presentation of the methods, as if the chapters were autonomous. The article would gain consistency if the introduction ended with a conclusion about the literature review carried out and the extent to which this helped the authors define the research focus.
Response 2:
Thank you so much for the observation. We re-structured the “Introduction” and connect the state-of-art with the objectives of the study. You can find in the page 6 in the manuscript in red colour.
Comment 3:
In the same vein, the chapter dedicated to presenting the methodology would benefit from a sentence summarising the data collection options before presenting the sample and the data collection platform used.
Response 3:
Thank you so much for the suggestion. We re-write the methodology section. You can find in the pages 6 and 7 in blue colour.
Comment 4:
The only thing that isn't clear is what criteria were behind the motivations presented (multiple choice) and why a criterion mentioned in the introduction doesn't appear: the search for well-being.
Response 4:
Thank you so much for your observation. In the pre-pilot study, we conducted a discussion group on various items for this pilot study, but no one specifically mentioned well-being. This may be because people inherently associate dietary care with overall well-being and health. We also tested the short questionnaire, and people referenced intoxication in the past, so we included this option.
Reviewer 2 Report
Comments and Suggestions for Authors
Dear Authors,
Thanks for the study, which aims to assess the motivations and channels for verifying nutritional information in Spanish university communities.
The authors have done a good literature review in the introduction and the discussion is extensive, but there are some uncertainties/questions about the study itself.
- The study involves three large universities, but the number of respondents is only 124. Is that enough to draw conclusions and to write paper? Given that this is a survey and not a clinical trial, the number of respondents should be much higher.
- The study does not provide any characteristics of the respondents, although the authors themselves describe in both the introduction and the discussion that eating habits/choices are significantly influenced by the demographic, cultural and contextual profile of the consumer.
- The study itself, which is included in the publication, is based on only five questions.
- Table 1 shows that only 54 respondents cross-check information on nutrition and food, which raises the question: Why do Tables 2 and 3 include responses from respondents who do not verify the accuracy of the information?
- Question about the percentages given in Tables 2 and 3. What is the reason for this, what does it explain?
- Recommendation to separate the discussion and conclusion parts.
- In the discussion section, lines 434 and 435, this was not reflected in the study, so such statements cannot be made.
- In the discussion part, lines 445 and 446, this was not reflected in the study (food labelling systems), so such statements cannot be written.
- In the discussion section, lines 453-455, authors are written about university students, faculty and administrative staff, who are not analyzed in the study, so such statements cannot be written.
Author Response
The responses could be found in the manuscript written in blue colour
Comment 1:
The study involves three large universities, but the number of respondents is only 124. Is that enough to draw conclusions and to write paper? Given that this is a survey and not a clinical trial, the number of respondents should be much higher.
Response 1:
We appreciate the reviewer’s concerns about our sample size of 124 respondents from three large universities. While this number may seem small, our study was intentionally designed as a pilot to explore how interested university communities are in verifying food and nutrition information. Pilot studies typically use smaller sample sizes since they serve as an early step in shaping larger-scale research. Given our study’s exploratory nature, this sample was appropriate for gaining initial insights. Our goal at this stage wasn’t broad generalizability but rather assessing the potential for scaling the research nationally.
We will make sure to clearly explain our sample size rationale and study objectives in the manuscript to highlight its role within the larger research framework. Also, we will add the reference was used.
Regarding the number of responses, we initially aimed for 100, which we determined to be sufficient for identifying trends and informing future research. However, participation exceeded expectations, and we included the additional responses to further enrich our preliminary findings.
The revised manuscript this change can be found – page 6, paragraph 4, and lines 1-4.
Comment 2:
The study does not provide any characteristics of the respondents, although the authors themselves describe in both the introduction and the discussion that eating habits/choices are significantly influenced by the demographic, cultural and contextual profile of the consumer.
Response 2:
We appreciate the reviewer's observation regarding our study's absence of socio-demographic characteristics. However, as this was a pilot study, our primary objective was to explore whether individuals actively verify food and nutrition information and the channels they use rather than analysing the influence of demographic factors on eating habits. At this early stage, we focused on assessing engagement with information verification to determine the feasibility of scaling the research. Based on the insights gained here, future studies will incorporate a broader sample and demographic variables to further explore these relationships.
Comment 3:
The study itself, which is included in the publication, is based on only five questions.
Response 3:
Thank you for your feedback regarding the number of questions in our study. Since this was a pilot study, we primarily aimed to examine whether people verify food and nutrition information, and which channels they use rather than conducting a detailed analysis at this stage. We carefully designed five key questions to keep the survey clear and focused, ensuring we could gather meaningful insights without overwhelming participants. This approach also allowed us to test the feasibility of our research as a proof-of-concept project, which helped us refine our methodology before expanding the proposal. By starting small, we could assess how people engaged with the survey, how they responded, and whether the topic resonated with them.
Comment 4:
Table 1 shows that only 54 respondents cross-check information on nutrition and food, which raises the question: Why do Tables 2 and 3 include responses from respondents who do not verify the accuracy of the information?
Response 4:
Maybe there is a misunderstanding, tables 2 and 3 show 97 respondents who are interesting in verifying the information related to nutrition and food.
Comment 5:
Question about the percentages given in Tables 2 and 3. What is the reason for this, what does it explain?
Response 5:
Regarding the 97 respondents who expressed an interest in verifying information, our aim was to understand the motivations behind their fact-checking behaviour as well as the platforms, media, and channels they rely on for this process.
Comment 6:
Recommendation to separate the discussion and conclusion parts.
Response 6:
Thank you for your suggestion! The changes have been made to the manuscript, with the discussion and conclusion sections now separated.
Comment 7:
In the discussion section, lines 434 and 435, this was not reflected in the study, so such statements cannot be made.
Response 7:
Thank you so much for your observation. In the discussion section, we interpreted the key findings of this pilot study and existing literature. It begins with a summary of the main results, followed by an explanation of their meaning and significance. So, we need to know what statements you are referencing.
Comment 8:
In the discussion part, lines 445 and 446, this was not reflected in the study (food labelling systems), so such statements cannot be written.
Response 8:
Thank you so much for your observation. In the discussion section, we interpreted the key findings of this pilot study and existing literature. It begins with a summary of the main results, followed by an explanation of their meaning and significance. So, we need to know what statements you are referencing.
Comment 9:
In the discussion section, lines 453-455, authors are written about university students, faculty and administrative staff, who are not analysed in the study, so such statements cannot be written.
Response 9:
Thank you for your suggestion! The abstract now includes the concept of the university community, encompassing students, faculty, and administrative staff. The survey was conducted during outreach activities at three universities, with respondents from various positions.
Round 2
Reviewer 2 Report
Comments and Suggestions for Authors
Dear authors,
thank you for the improved version of the manuscript. I hope that this pilot study will be continued in nationwide study.
Comments on the Quality of English LanguageIt is not in my expertise.